# Targeting Granulin Haploinsufficiency in Frontotemporal Dementia: From Genetic Mechanisms to Therapeutics

**DOI:** 10.3390/ijms26209960

**Published:** 2025-10-13

**Authors:** Eva Bagyinszky, Seong Soo A. An

**Affiliations:** 1Graduate School of Environment Department of Industrial and Environmental Engineering, Gachon University, Seongnam 13120, Republic of Korea; eva85@gachon.ac.kr; 2Department of Bionano Technology, Gachon Medical Research Institute, Gachon University, Seongnam 13120, Republic of Korea

**Keywords:** frontotemporal dementia, haploinsufficiency, granulin, progranulin, mutation

## Abstract

Frontotemporal dementia (FTD) is the second most common early-onset dementia after Alzheimer’s disease, characterized by progressive neurodegeneration primarily in the frontal and temporal lobes. *Granulin* (*GRN*) gene for encoding the progranulin (PGRN) protein was a key genetic contributor to FTD. PGRN was a multifunctional protein involved in lysosomal function, neuroinflammation, and neuronal survival. This review discusses the contributions of *GRN* haploinsufficiency to FTD pathogenesis with an emphasis on genetic mutations, downstream cellular consequences, relevant animal and cellular models, and emerging therapeutic strategies. Loss-of-function mutations in *GRN* were responsible up to ~50% reduction in PGRN levels, resulting in lysosomal dysfunction, TDP-43 aggregation, impaired microglial homeostasis, and enhanced neuroinflammation. Multiple in vitro and in vivo models recapitulated these pathological features. Novel therapeutic approaches, such as AAV-mediated gene therapy, stop codon readthrough compounds, SORT1 inhibitors, and antisense oligonucleotides, were investigated to restore PGRN levels and to mitigate disease progressions. However, challenges included the oncogenic risks of overexpression and the limited translational success in clinical trials to date. Targeting *GRN* haploinsufficiency became a promising avenue for FTD therapy. Improved models and refined delivery systems would be essential to develop safe and effective treatments. Future work should also focus on biomarker-guided interventions in presymptomatic mutation carriers.

## 1. Introduction: Structure and Functions of Granulin Gene and Protein

Frontotemporal dementia (FTD) is the second most common form of dementia after Alzheimer’s disease (AD). FTD is a complex group of neurodegenerative disorders, which could be characterized by atrophy in the frontal and temporal lobes with diverse clinical presentations and multiple genetic causes [1]. The main clinical subtypes of FTD included the behavioral variant (bvFTD), semantic dementia (SD), asemantic variant primary progressive aphasia (svPPA), and non-fluent variant primary progressive aphasia (nfvPPA) with additional phenotypes, such as right temporal variant FTD and FTD overlapping with motor neuron disease (FTD-MND) [1,2,3,4,5]. Up to 50% of ALS patients developed cognitive impairment with features of FTD, and approximately 30% of FTD patients exhibited the motor dysfunction characteristics of ALS, indicating a broad neurodegenerative spectrum [5,6].

Given its complexity, numerous genetic factors were identified as causative or risk factors for FTD. Repeat expansions in *C9orf72*, and mutations in Progranulin (PGRN) and Microtubule-Associated Protein Tau (*MAPT*) were the main FTD-causing genetic factors. Recently, additional identified genetic factors of FTD were Valosin-containing protein (*VCP*), Charged Multivesicular Body Protein 2B (*CHMP2B*), TAR DNA Binding Protein (*TARDBP*), FUS RNA Binding Protein (*FUS*), Sequestosome-1 (*SQSTM1*), Coiled-Coil-Helix-Coiled-Coil-Helix Domain Containing 10 (CHCHD10), TANK-binding kinase 1 (*TBK1*), optineurin (*OPTN*), Cyclin F (*CCNF*), and T-cell intracellular antigen 1 (*TIA1*) genes [7,8,9].

Among the above, the *granulin* gene (*GRN*), which encodes the PGRN, was a key genetic contributor to FTD. *GRN* on chromosome 17q21.31 (between 44,345,246 and 44,353,106 on the hg38 genome) contained 12 coding and one non-coding exon. PGRN protein, a 593-amino acid-long glycoprotein, consists of seven (P-G-F-B-A-C-D-E) and a half tandem repeats of granulin/epithelin module (GEM). Among the repeats, the P motif was the half domain, and the rest contained the full repeats. The individual PGRN domains are around 60 amino acids long (Figure 1) with 12 conserved cysteine residues for stabilizing the β-hairpins, global structure, and the overall protein structure [10,11,12].

PGRN protein was processed by various intracellular and extracellular protease enzymes, in which the cleavages could occur in both extracellular (in the extracellular matrix) and intracellular areas, particularly in the lysosomes [8,9,10,11,12,13]. In the extracellular space, PGRN elastases, extracellular proteases (such as proteinase 3), and matrix metallopeptidases (MMPs) made their specific cleavages [8,9,10,11,12,13]. Inside the lysosomes, PGRN is processed by proteases into individual GRN peptides through the action of different enzymes, including cathepsin L or cysteine proteases [14,15]. Lysosomal PGRN trafficking was regulated by two different pathways. The first involved PGRN’s interaction with sortilin (SORT1) protein, while the second was mediated by prosaposin (PSAP) through mannose-6-phosphate receptors (M6PR) and lipoprotein receptor-related (LRP1) proteins. Full-length PGRN was also located in the compartments of secretory pathways and moved towards the extracellular spaces through exocytosis (Figure 1b). The processing of PGRN through lysosomes plays an essential role in balancing inflammatory pathways [10,16,17]. The PGRN levels could be modified by different factors, including SORT1, PSAP, or receptor-interacting serine/threonine protein kinase 1 (RIPK1) proteins [18].

Several functions of PGRN in various pathways were identified, which involved a growth factor, promoting cell proliferation, survival, and migration. PGRN, as a mitogenesis inducer in epithelial cells, was implicated in tumor progression and transformation by inducing cyclin D1 expression and activating various mitogenesis and survival pathways (ERK or PI3K/Akt). As an oncogene, PGRN demonstrated its role in different cancers, including ovarian, breast, and hepatic carcinoma [19,20]. PGRN expressions were upregulated at wound sites and were verified to impact tissue recovery, matrix remodeling, and immune modulation during injury healing [21,22]. The full-length PGRN was revealed to contain anti-inflammatory effects by inhibiting pro-inflammatory cytokines. Furthermore, PGRN, as a neurotrophic factor, protected neurons against excessive microglial activation and neuroinflammation. However, after protease cleavages, the GRN peptides may promote pro-inflammatory responses and attract immune cells to injury sites [23]. In the brain, PGRN promoted neuroprotection, survival of nerve cells, neurite outgrowth, and synaptic pruning [8,24,25,26,27,28,29]. PGRN was also verified to be essential for the homeostasis of lysosomal functions, such as protein degradation and lysosome biogenesis. PGRN regulated the lysosomal acidification and protease activities through cathepsin D or prosaposin-mediated sphingolipid hydrolysis [23,30]. Figure 2 summarizes the diverse functions of the PGRN protein.

Loss-of-function mutations in the *GRN* gene were considered as causative factors for neurodegenerative diseases, like FTD and lysosomal storage disorders [31,32]. Pathogenic mutations in the *GRN* gene were reported in approximately 5–10% of FTD patients, but they could be more frequent (5~20%) in the case of familial FTD [31,32,33,34]. This manuscript presents the role of haploinsufficiency in association with *GRN* mutations in FTD. The relevant animal and cell models were discussed, and the potential strategies for targeting *GRN* haploinsufficiency were explored.

## 2. GRN Mutations, FTD, and Haploinsufficiency

As previously mentioned, *GRN* mutations were considered as a primary disease-causing factor in FTD from an inherited autosomal dominant manner [33,34]. Various mutations in *GRN* included the STOP codon, splice site, and frameshift mutations, potentially leading to premature termination of *GRN* mRNA and protein truncations [18,26,35,36,37,38,39,40]. Mutations of the START codon (ATG or AUG in mRNA) cause the failure of its translation initiation [41,42]. Furthermore, missense mutations, such as Ala9Asp or Trp7Arg at the signal peptide, cause a loss-of-function effect by disrupting the signal recognition particle (SRP), resulting in a consequential reduction in GRN expression [43,44]. Beyond coding regions, variants in the 5′ region of *GRN* influenced its expression regulations or mRNA stability, potentially leading to increased or reduced PGRN protein levels [42]. Additionally, the common intronic variant, rs5848 (g.12754C > T), was correlated to PGRN levels, and the TT allele was suggested to reduce serum PGRN levels, acting as a risk factor for FTD and other dementias [45].

*GRN* mutation carriers typically did not exhibit Tau aggregates; instead, common findings were ubiquitin-immunoreactive neuronal inclusions and TDP43 aggregates. *GRN* mutations caused a broad spectrum of clinical phenotypes in FTD patients. While behavioral variant FTD (bvFTD) was the most frequently identified phenotype, several patients with *GRN* mutations were also diagnosed with primary progressive aphasia (PPA), corticobasal degeneration syndrome (CBDS), Lewy body dementia (LBD), or AD-like symptoms. Additional atypical symptoms, such as early Parkinsonism, visual hallucinations, motor apraxia, or episodic memory impairment (hippocampal amnestic syndrome), were also observed among its mutation carriers [41,46,47,48]. Brain imaging in mutation carriers often revealed hypoperfusion in the frontotemporal regions, although other brain areas, such as the hippocampus, parietal lobe, and posterior cingulate gyrus, were also affected [41]. The age of onset varied significantly, with cases identified from the 40s and after 80 years of age [49,50]. Phenotypic variability even occurred within the same family, suggesting that other genetic-epigenetic or environmental factors could modulate disease presentation [50,51].

Interestingly, *GRN* mutations (e.g., Cys139Arg, Pro451Leu, or rs5848) were also reported in AD patients, which suggested that *GRN* could be a potential risk factor for AD. Specifically, the rs5848 AA allele was found to reduce the PGRN levels in the brain and plasma of AD patients [8,45]. However, further studies should be performed on the exact AD-related mechanisms of *GRN* mutation, as its reduced expression has been reported to be beneficial in cases of amyloid plaques. However, *GRN* may be involved in AD progression through amyloid-independent pathways [24]. Table 1 provides examples of GRN mutations with their effect of haploinsufficiency (reduced mRNA and protein levels). Figure 3 summarizes the location of *GRN* mutations and their location in the *GRN* gene and the PGRN protein.

## 3. Biomarkers of *GRN* Haploinsufficiency

As previously mentioned above, the majority of *GRN* mutations were associated with loss-of-function mechanisms, leading to reduced plasma PGRN levels in affected patients. Analyzing plasma PGRN in novel *GRN* carriers became an effective method to evaluate the pathogenicity of its mutations. Moreover, monitoring PGRN levels in biological fluids could be a valuable approach for assessing the efficacy of drug candidates that target *GRN* haploinsufficiency [79,80,81]. Circulating PGRN levels were reduced in serum or CSF in cases of mutations in association with its haploinsufficiency [73,82,83]. Among these biomarkers, monitoring plasma PGRN levels would be crucial for identifying individuals with probable pathogenic *GRN* mutations, as reduced plasma progranulin levels reflect the loss-of-function mechanisms [79,80,81]. Furthermore, plasma PGRN levels have been suggested as a potential biomarker for predicting disease progression in presymptomatic *GRN* mutation carriers [80]. They may also be important in monitoring potential drug candidates for *GRN* haploinsufficiency, as successful candidates may increase and stabilize plasma granulin levels [79,80,81,82,83].

Another promising biomarker for *GRN* haploinsufficiency was the neurofilament light chain (NfL) protein, which was considered an accurate marker for both neuroinflammatory and neurodegenerative changes. The elevated levels of NfL could be useful in the early detection of GRN-haploinsufficiency-related neurodegenerative damage [84]. Additionally, CSF NfL levels correlated with disease severity and with hypometabolism in affected brain regions [85]. Hence, CSF NfL could be useful in monitoring the FTD drug candidates, since their reduced levels may be associated with a positive therapeutic outcome, resulting in a slowdown in neurodegeneration [84,85]

Plasma analysis of patients with *GRN* haploinsufficiency-associated mutations also revealed elevated levels of peripheral biomarkers related to macrophage-mediated innate immunity, such as plasma sCD163 and CCL18, which were higher in GRN mutation carriers than in controls. Plasma LBP may serve as a useful marker for disease severity in *GRN* mutation carriers, as its levels correlated with frontal white matter integrity [86]. A separate study indicated that FTD patients with *GRN* haploinsufficiency had higher serum concentrations of C-peptide and resistin. Another biomarker, Ghrelin, was found to be increased in both presymptomatic *GRN* mutation carriers and FTD patients, suggesting that metabolic biomarker alterations may be associated with FTD progression [87]. Research into the impact of *GRN* haploinsufficiency on brain sphingolipid enzymes revealed the reduced beta-glucocerebrosidase activity in the inferior frontal gyrus of *GRN* carrier FTD patients. Specifically, mature enzyme levels were lower, and insoluble and glycosylated beta-glucocerebrosidase was activated in their neurons. These findings, corroborated in *GRN* knockout mice, suggested that *GRN* haploinsufficiency may be linked to impaired beta-glucocerebrosidase processing [88]. Elevated peripheral inflammatory markers (e.g., sCD163, CCL18) and metabolic biomarkers (e.g., C-peptide, resistin, ghrelin) in *GRN* carriers could be useful markers for monitoring disease severity and metabolic alterations, potentially predicting a more aggressive disease course. This suggested a broader panel of markers for better-informed prognostic evaluations. Changes in the levels of peripheral inflammatory markers or metabolic markers may also be useful in monitoring the efficacy of therapeutic candidates for *GRN* haploinsufficiency [86,87,88].

## 4. Patient-Derived Cell Lines of *GRN* Haploinsufficiency

FTD patient-derived cell lines with various *GRN* mutations revealed several impairments, including disrupted TDP43 metabolism, inflammatory dysfunctions, lysosomal abnormalities, and reduced cell survival (Figure 4). As mentioned before, patients with pathogenic FTD mutations could present diverse phenotypes, even in cases of the same mutations [41,42,43,44,45,46,47,48,49,50,51,52,53,54,55,56,57,58,59,60,61,62,63,64,65,66,67,68,69,70,71,72,73,74,75,76,77,78]. However, in the case of patient-derived cell lines, the effects of *GRN* mutation presented uniformity. The reason for diversity in the patients could be complex, including genetic or epigenetic factors, and the presence of environmental influences [51]. However, patient-derived cell lines were studied in a controlled environment, since these studies may focus more on the effects of *GRN* haploinsufficiency [89,90,91,92]. Specifically, patient-derived iPSC cells from TDP43-positive individuals with *GRN* haploinsufficiency exhibited reduced nuclear TDP43 levels; however, insoluble TDP43 levels were elevated. Furthermore, *GRN* mutant cells were associated with abnormal lysosomal functions and neuronal ceroid lipofuscinosis (NCL), characterized by reduced proteolysis and decreased expression of cathepsin D in cortical neurons. A study identified full-length PGRN and GRN E as activators of cathepsin D, but not cathepsin B or L. The diminished cathepsin D activity due to reduced PGRN expression contributed to FTD phenotypes and NCL pathology in *GRN* haploinsufficiency [89]. Additionally, PGRN was responsible for the neuronal uptake and lysosomal delivery of prosaposin (PSAP), a precursor to saposins involved in glycosphingolipid degradation. *GRN* haploinsufficiency may result in low PSAP levels in mice and human-derived cell lines [90]. In *GRN* haploinsufficient iPSCs from patients with the *GRN* mutation, Wnt signaling was upregulated, inhibiting cortical neuron generation, a defect restored by stimulating normal PGRN expression [91]. Multiple cell models for Ser116Ter *GRN* mutations were developed. iPSC cells with Ser116Ter exhibited increased sensitivity and reduced viability when exposed to various kinase inhibitors, including those targeting PI3K/Akt or MEK/MAPK signaling. The results suggested that the reduced *GRN* expressions were associated with lower levels of pro-survival factors and heightened sensitivity to environmental stressors [92]. Treatment of iPSC or Neuro-2a cell lines from Ser116Ter carrier FTD patients with histone deacetylase inhibitors, such as superpolyamide hydroxamic acid (SAHA), caused the elevated PGRN production in cortical neurons. While SAHA treatment altered the gene expression profiles, it did not improve the sensitivity of Ser116Ter carrier cell lines to stress-inducing signals, warranting further studies on SAHA’s side effects [93,94]. Human iPSC (hiPSC) cell lines generated from Portuguese patients carrying a homozygous or heterozygous frameshift mutation (Ser301Cysfs61) in GRN displayed varied PGRN levels. While one heterozygous patient’s cell line showed the expected 50% reduction in PGRN levels, another carrier, who was in a preclinical phase and asymptomatic at the time of analysis, exhibited higher PGRN levels than controls, emphasizing the complexity of PGRN expression in early disease stages. This study also generated brain organoids from patient cell lines with embryoid bodies (EBs) from homozygous and heterozygous frameshift mutation carriers, showing the reduced size, but successful development into human brain organoids. All cell lines (normal, homo-, and heterozygous *GRN* carriers) expressed ectodermal development markers (e.g., SMA, GATA4, NESTIN, and βIII-TUBULIN). *GRN* gene and PGRN protein expressions in whole brain organoids were similar to those in hiPSCs, suggesting these cell lines hold promise for studying FTD’s pathogenic mechanisms [95].

Derived fibroblasts from patients with *GRN* haploinsufficiency presented various disease-associated changes, including higher levels of insoluble p62, indicating autophagy dysfunction. A study revealed that *SORT1* knockout cells with *GRN* haploinsufficiency showed the increased *GRN* expressions, suggesting that the *SORT1-GRN* axis influenced the lysosomal function regulation. TMEM106B stimulation elevated the full-length PGRN expression within cells, but reduced 2,3GRN levels, implying that TMEM106B might inhibit PGRN processing into GRNs. TMEM106B could also inhibit PGRN through abnormal lysosomal dysfunctions and impaired trafficking with *GRN* haploinsufficiency, potentially initiating lysosomal impairment. *GRN* deficiency is revealed to cause abnormal activity of lysosomal proteases and accumulation of NCL-like materials in the cortex [17,96,97].

Monocyte-derived microglia (iMGs) from FTD patients with pathogenic or probable pathogenic *GRN* mutations (Met1Val and Trp147Ter) exhibited significant neurodegenerations. Plasma samples from *GRN* mutation carriers showed reduced plasma PGRN protein levels, and transcriptome analysis revealed reduced PGRN mRNA. Biomarkers of neuroinflammation (sTREM2 and NfL) were also elevated in the CSF of these patients. Analysis of inflammation-associated genes revealed the reduced expressions of genes in microglial homeostasis (including P2RY12, TMEM119, TGFBR1, and CX3CR1), while pro-inflammatory genes, like IL1beta, TNFα, and IL6, were elevated in iMGs from patients, suggesting impaired phagocytic pathways and accelerated neuroinflammations. These cell models also displayed cytoplasmic TDP43 inclusions, which were granular, dot-like, or round, and contained phosphorylated TDP43 with ubiquitin, alongside the reduced phagocytosis. Furthermore, iMGs with *GRN* haploinsufficiency exhibited abnormal lysosomal functions and impaired lipid metabolism, potentially leading to further impairment in the phagocytic clearance of protein aggregates [84]. *GRN* gene deficiency may lead to gangliosidosis due to impaired lysosomal lipid degradation, contributing to neuroinflammation and neurodegeneration. *GRN* deficiency could result in impaired lipid metabolism, particularly affecting bis(monoacylglycerol)phosphate (BMP), which was crucial for ganglioside degradation and accumulation [98].

Peripheral cells (lymphoblasts) isolated from *GRN* mutation-carrier (c.709-1G > A) with TDP43-positive FTD patients showed that *GRN* haploinsufficiency stimulated NFkappaB signaling and overactivated Wnt5a signaling, leading to increased intracellular Wnt5a levels and Wnt5a secretion. This study proposed that the Wnt5a-related cell cycle could be a potential indicator of disease progression, and Wnt5a may serve as a peripheral marker for TDP43-positive FTD, emphasizing its significant role in disease progression [99]. The increased Wnt5A expressions were attributed to the expression of TNFα and NFkappaB signaling [100]. Zhu et al. (2019) [101] developed neuroblastoma cell lines with *GRN* haploinsufficiency using short hairpin RNA, which displayed TDP43-positive inclusions, hyperphosphorylation, and abnormal C-terminal cleavage of TDP43. Normal TDP43 distribution was restored upon inducing PGRN expression [101].

Taken together, patient-derived cell lines with various GRN mutations demonstrated different impairments, including TDP43 metabolism, inflammatory dysfunctions, lysosomal abnormalities, or reduced cell survival. The benefits of these models could be that they could offer a direct human-specific context for understanding GRN haploinsufficiency, providing insights that may be challenging to obtain from animal models [90,91,92,93,94,95,96,97,98,99,100,101,102,103,104,105,106,107,108,109,110]. Cell models should be essential, since identifying the cellular impairments in the case of *GRN* haploinsufficiency could provide more opportunities in the case of drug development and testing. Drugs specifically designed for human physiology can be initially tested in these cell lines. Cell lines could be beneficial in understanding the disease heterogeneity by comparing the responses of the cells from patients with the same *GRN* mutations. This is particularly important in the development of personalized medicine, as understanding the differences in disease progression and response to treatment may be unique, and individualized therapeutic strategies may be necessary. Furthermore, patient-derived cell lines could provide a scalable and ethical platform for high-throughput screening of therapeutic compounds and candidates, before testing them in animal models or patients. The cell lines could also be beneficial for biomarker discovery in cases of granulin haploinsufficiency (e.g., WNT5A), which may aid in disease diagnosis and monitoring disease progression [90,91,92,93,94,95,96,97,98,99,100,101]. The disadvantage of cell lines is that they operate in an isolated environment, focusing solely on specific cellular effects. Furthermore, the gene-environmental interactions may be difficult to model in the cell lines [90,91,92,93,94,95,96,97,98,99,100,101].

## 5. Animal Models of *GRN* Haploinsufficiency

Several mouse models were employed to investigate *GRN* deficiency. Heterozygous *GRN* knockout mice exhibited age-related emotional and social impairments in their behavior, although no gliosis, TDP43 pathology, or lipofuscinosis were detected. These mice also displayed reduced neuronal activation in the amygdala [102]. *GRN* knockout mice generally showed increased pro-inflammatory signal production and abnormal behavior. Specifically, heterozygous *GRN* knockout mice presented the impaired biphasic social dominance, initially being more dominant than wild-type mice at 6–8 months of age, with this dominance subsequently decreasing. Between 6 and 9 months, these mice demonstrated increased mTORC2/Akt signaling in the amygdala, accompanied by enhanced dendritic arbors in this region. However, after 9 months, basal dendritic arbors were reduced in the prelimbic cortex [103]. In *GRN* haploinsufficient mice, changes in dendritic morphology were observed, particularly in the frontal cortex, including an altered apical dendrite ratio, reduced numbers of stubby spines, increased thin spines, longer spines, and smaller head diameters. This abnormal dendritic spine morphology caused the circuit impairments, such as in the MD-mPFC circuit, resulting in an altered excitation/inhibition ratio and abnormal social behavior [104]. Humanized mouse models with heterozygous *GRN* knockout exhibited behavioral dysfunctions, including hyperactivity, repetitive behaviors (e.g., marble burying), and anxiety. Both homozygous and heterozygous *GRN* knockout mice developed microgliosis by 18 months of age; however, no exaggerated lipofuscinosis or TDP-43 aggregates were detected. Furthermore, both homo- and heterozygous *GRN* knockouts could alter gene expressions in the brain; at 18 months, 918 differentially expressed genes were identified in heterozygous *GRN* knockout mice than wild-type controls [105].

A mouse model with a homozygous *GRN* Arg493Ter mutation displayed distinct pathological hallmarks, including impaired lysosomal functions and degeneration of specific thalamic regions. Inflammatory phenotypes, such as neuroinflammation and astrogliosis, were also observed in the hippocampus and thalamus of these mutant mice, alongside notable lysosomal expansions. Interestingly, inhibitory synaptic density was partially preserved in mice with the Arg493Ter mutation. These mice exhibited behavioral changes, including anxiety, social dominance, and excessive grooming, similar to those observed in Grn−/− mice. Homozygous Arg493Ter mice showed increased TDP-43 phosphorylation, unlike heterozygous mice. However, both homo- and heterozygous mice presented the stimulated expressions of lysosomal genes and pro-inflammatory molecules with increased microgliosis and astrogliosis. Mouse models with *GRN* Arg493Ter were promising for studying therapeutic approaches for *GRN*-related FTD [51,106,107,108]. Heterozygous *GRN* knockout mice had lower PGRN levels in their interstitial fluid and brain tissue (~50% reduction), particularly in the cortex, compared to wild-type mice [109].

Mouse models (APP/Grn+/−) demonstrated that amyloid deposition was significantly reduced in mice with *GRN* haploinsufficiency than normal *GRN*. However, *GRN* haploinsufficiency could lead to other dysfunctions, including neuroinflammation or Tau pathology-related pathways [110,111,112]. Studies in mice with heterozygous knockout of both *TMEM106B* and *GRN* revealed that *TMEM106B* reduction only offered minor beneficial effects in *GRN* haploinsufficiency, such as restoring beta-glucuronidase activity and improving lysosomal function. However, these mice did not exhibit the improved social behavior [113]. Mice with heterozygous knockout of *GRN* and a *MAPT* Pro301Leu mutation showed increased Tau phosphorylations and elevated cyclin-dependent kinase activities, suggesting that *GRN* haploinsufficiency impacted Tau phosphorylations and accumulations [114]. In addition, mice with TREM2 deficiency and *GRN* haploinsufficiency were associated with lower microglial hyperactivations [115].

Hippocampal cultures from E18 rat embryo cell lines, where *GRN* levels were knocked down, showed reduced neural connectivity and synaptic density, as well as decreased neural arborization and length. Conversely, neural transmission was increased, including more synaptic vesicles per synapse and a higher frequency of spontaneous glutamatergic transmission. Postmortem brain samples from FTD patients with *GRN* mutations showed similar results, suggesting that increased presynaptic release might be a defensive mechanism to maintain synaptic communication [116]. Neuronal cultures from C57BL/6J mice with reduced *GRN* expression exhibited decreased levels and density of GluN2B-containing N-methyl-D-aspartate (NMDA) receptors and reduced NMDA-dependent Tau phosphorylation, which were associated processes with the inhibition of neuronal arborizations, key factors in neural structural plasticity [117].

Studies in *GRN* haploinsufficiency mouse models indicated that SORT1 knockout increased *GRN* expression, though increased *GRN* expression did not affect SORT1 levels in the mouse brain [118].

In summary, animal models, particularly mouse models, were designed to understand the in vivo effects of *GRN* haploinsufficiency, as they offer an opportunity to study the complex neural circuits, behavioral phenotypes, and long-term pathological progression that may be difficult to replicate in cell systems [102,103,104,105,106,107,108,109,110,111,112,113,114,115,116,117,118]. Mice with *GRN* haploinsufficiency may exhibit behavioral issues and circuit impairments in their nerve cells. Also, they presented microgliosis, neuroinflammation, or lysosomal dysfunctions [102,103,104,105,106,107,108,109,110,111,112,113,114,115,116,117,118]. Mouse models may offer advantages over animal models in that they represent the complex, multi-systemic nature of FTD, including behavioral deficits, neuronal circuit dysfunction, and age-related progression of pathology. Animal models could also be essential in testing the therapeutic candidates of FTD, e.g., their safety and their ability to penetrate the brain, possible side effects, or their response to therapies, before moving them to animal models. Also, animal models could be important in the discovery and validation of biomarkers. Furthermore, animal models may be more effective in modeling and understanding gene interactions between *GRN* and other genes (e.g., with *SORT1* or *TREM2*) and their role in disease progression. Additionally, studies in animal models may be useful in investigating the compensatory mechanisms that could occur in response to GRN deficiency, including increased presynaptic release in hippocampal cultures, which may explain the delayed onset or various symptoms in patients [102,103,104,105,106,107,108,109,110,111,112,113,114,115,116,117,118]. However, the disadvantages of animal models could be that they may not exhibit the key neuropathological features of FTD, e.g., TDP43 aggregates. Animal models may not be effective in modeling the human FTD-related disease mechanisms. There may be significant differences between the physiology and anatomy of animals and humans, which may result in mismatches between the data from human-derived cell lines and animal models. Ethical concerns may also be an issue in the case of animal models [102,103,104,105,106,107,108,109,110,111,112,113,114,115,116,117,118].

## 6. Possible Therapeutic Targeting of *GRN* Haploinsufficiency

Currently, there is no therapy available for FTD. Targeting the *GRN* gene may be a promising approach in FTD therapy; however, other studies examined the side effects and the risk-benefit ratio in *GRN*-related therapy [24,119,120,121,122,123,124,125,126,127,128,129,130,131,132,133,134,135,136,137,138,139,140,141,142,143,144,145,146,147,148,149,150,151,152,153,154,155,156,157,158,159,160,161,162,163,164,165,166,167,168,169,170,171,172,173,174,175,176,177,178,179,180,181,182,183,184,185,186,187,188,189,190,191,192,193,194,195,196,197,198,199]. Gene therapy approaches were ongoing, which focused on delivering the *GRN* gene to increase the progranulin levels. Viral vectors, such as adeno-associated virus (AAV), were a useful method to stimulate GRN expression in the case of *GRN* deficiency. Mice with granulin knockout presented milder symptoms (e.g., microgliosis and lipofuscinosis) after AAV-mediated expression of *GRN*. After *GRN* stimulation, the microglial homeostasis improved the dysregulated lysosomal proteins [124]. PGRN AAV gene therapy (PR006) was initially tested in preclinical trials using *GRN* haploinsufficient mice. After three months of PR006 treatment, mice showed elevated *GRN* expressions and reduced expressions of pro-inflammatory proteins, along with decreased markers for microgliosis and astrogliosis. PR006 was further tested in non-human primates, suggesting that this gene therapy could be a safe therapeutic target in humans. In Phase 1–2 clinical trials, the drug was well tolerated, as no anti-PGRN antibodies were observed in the CSF. PR006 administration led to a transient increase in PGRN levels in the CSF and plasma of patients. After dosing, NFL levels were initially increased but returned to baseline after 9–12 months. Bis(monoacylglycero)phosphate (BMP) phospholipids also showed increased levels in patients’ urine. However, there was no conclusive proof yet whether PR006 could slow down the disease progression, and further studies were planned, particularly in patients in the early stages of FTD [122,124,125]. The AVB-101 was also an AAV (AAV9)-based therapy, which started its clinical Trial1/2 in individuals with *GRN* mutations and FTD therapy. However, no information was available on its effect on PGRN levels or safety. Preclinical studies in primates revealed the increased PGRN levels in the brain, cardiovascular system, and liver after AVB-101 treatment [153,166,167]. PBFT02 was also an AAV-based gene therapy, currently in Phase 1b. The initial results of PBFT02 appeared promising, as the tested patients exhibited elevated CSF progranulin levels at 6 months following the initiation of treatment. Also, the therapy seemed to be well tolerated in patients [168,169,170]. The benefit of AAV vector-related gene therapies was the long-term expression in the neurons. Preclinical data and early clinical data presented the elevated gene expressions. In addition, AAV-based methods facilitated targeting specific cell types in or brain regions, and they were well tolerated in preclinical and clinical studies. However, AAV-based therapies had several limitations. One of the issues was the invasive procedures of intracranial injections to efficiently target the brain. Another issue could be that AAV-induced immune responses may need immunosuppression procedures. Furthermore, the risk for off-target effects needed to be assessed [124,125].

Non-viral vectors, including lipid nanoparticles, may also be promising in gene delivery. To date, no study has been performed on *GRN* delivery by lipid nanoparticles, but it may be a safe and effective method in studying haploinsufficiency [171]. Additionally, the nanoparticles with enhanced stability and control over the release of drugs should be essential for drug delivery, especially targeting specific sites [172]. A thermo-sensitive hydrogel (Pluronic F-127) was successfully used to deliver progranulin and enhance the healing of corneal injury, which inhibited the inflammation (e.g., by suppressing NF-κB, PI3K/Akt, or Wnt/β-catenin signaling) and stimulated the axonal regeneration [173].

Besides gene therapies, epigenetic therapies may also be useful in targeting *GRN*. Brain tissues from FTD patients exhibited hypermethylation in certain regions of the *GRN* promoter with a significant inverse correlation between methylation and *GRN* mRNA levels, suggesting in vivo epigenetic regulation. The DNA methyltransferase DNMT3a was upregulated in the brains of FTD patients, and its overexpression was found to reduce *GRN* promoter activity, leading to decreased *GRN* expression in cell models. The DNA methyltransferase inhibitor 5-aza-2′-deoxycytidine (DAC) was shown to increase *GRN* expression in human lymphoblast cells and mouse microglia [126]. The Histone Deacetylase Inhibitor FRM-0334 was tested in randomized clinical trials of 27 patients with *GRN* mutations. While this drug appeared safe and well-tolerated, FRM-0334 failed to elevate plasma PGRN concentrations, possibly due to inconsistent absorption or poor oral bioavailability. Combining histone deacetylase inhibitors with GRN enhancers and other epigenetic target modulators may be promising for *GRN*-positive FTD [85,121,127]. SAHA (Vorinostat), a histone deacetylase inhibitor, was identified as a small molecule enhancer of *GRN*, increasing *GRN* gene and PGRN protein expressions in cultured cells and patient-derived cells with *GRN* haploinsufficiency, though it was not yet approved for FTD clinical trials [93]. Another epigenetic modification factor, a BET bromodomain inhibitor, was suggested to enhance *GRN* expression in neural cells from *GRN* mutation carriers [128].

Antisense oligonucleotides (ASOs) could also be promising in stimulating *GRN* expression. Studies on neuroglioma cells with *GRN* haploinsufficiency showed that blocking the miR-29b binding site with ASOs (such as M5, M10, or M36) increased the *GRN* expression [140]. Advantages of ASOs included their ability to increase the expression of the normal GRN allele and prevent abnormal GRN expression. Furthermore, targeting miR-29b could be used against any pathogenic *GRN* mutations. In the future, ASOs will be tested in FTD animal models to determine whether they can improve behavioral deficits and neuropathology. However, the repeated administration of ASOs may be needed to maintain the therapeutic levels. Even though ASP was safe, the other limitations are the risks for potential toxicity or off-target effects. Delivery of ASOs to the brain may also be challenging with intrathecal administration [140].

Studies were ongoing on drugs that could induce premature stop codon readthrough (PTC readthrough) in cases of *GRN* mutations, which may restore the normal, full-length PGRN protein. PTC readthrough was verified as a translational process in which the ribosome bypassed the premature stop codons, and it could induce the synthesis of full-length proteins despite the presence of nonsense mutations. Aminoglycosides, including G418 and gentamicin, were found to increase the tendency of PTC readthrough in combination with a phthalimide PTC readthrough enhancer. Cell models with Arg418Ter and Arg493Ter mutations were responding to G418 treatment, showing increased PGRN levels. Furthermore, *GRN* expressions were higher in Arg493Ter−/− KI hiPSC-derived neurons and astrocytes. The treatment also improved lysosomal homeostasis and function in these cell lines. While PTC readthrough therapies could be a promising direction for treating *GRN* haploinsufficiency, optimization would be needed to overcome clinical challenges, such as toxicity and bioavailability, suggesting the development of safer and more effective readthrough compounds [129,130].

Animal and cell studies suggested that the SORT1-PGRN axis may be a possible target for developing therapies for *GRN* haploinsufficiency-associated FTD. Studies on cell lines from FTD patients and mice also confirmed that inhibiting or knocking out SORT1 elevated the *GRN* expression in cases of *GRN* haploinsufficiency. Preclinical studies were currently underway on SORT1 antagonists, like 1-[2-(2-tert-butyl-5-methylphenoxy)-ethyl]-3-methylpiperidine (MPEP) and small-molecule PGRN-specific binders (PGRN C-terminal motif, PGRN (588–593). Treatment with MPEP in iPSCs (*GRN* Ser116Ter) and lymphoblastoid cell lines (Cys31Leufs*34 and Arg418Ter) with *GRN* haploinsufficiency was associated with lower SORT1 levels and increased PGRN levels. The C-terminal binding motif of the PGRN protein (588–593) was found to prevent SORT1-mediated endocytosis of the PGRN protein. This data suggested a neutrophil elastase site between Ala-588 and Leu-589, and disrupting this site could prevent PGRN cleavage and SORT1 interaction [116,136].

Amiodarone, a heart medication, was suggested to increase GRN expression and production through its possible effects on endosomal sorting in vitro [131]. However, a pilot study of amiodarone with five patients with a pathogenic *GRN* mutation did not have any beneficial effects on peripheral PGRN levels or disease progression [132]. Vacuolar ATPase inhibitors, including bafilomycin A1, concanamycin A, archazolid B, and apicularen A, along with alkalizing drugs such as chloroquine, bepridil, or amiodarone, were tested in mouse cell models for *GRN* haploinsufficiency. Among these, bafilomycin A1 was found to increase PGRN levels and secretion in the cells [133]. Bafilomycin A1 did not affect lysosomal and autophagosome degradation, suggesting it could increase *GRN* expression through independent pathways from lysosomal degradation. Furthermore, bafilomycin A1 could increase PGRN levels through transcription-independent mechanisms, as the *GRN* mRNA levels in the cell lines did not significantly change. However, acidosis was identified as the main side effect of balfomycin A1 treatment. Further studies would be needed on alkalizing reagents (such as NH4Cl and CQ) to prevent the acidosis caused by bafilomycin A1. This study suggested that vacuolar ATPase inhibitors may be potential future therapeutic agents for FTD [133].

Trehalose, a disaccharide found to activate autophagy, was demonstrated to increase PGRN levels in both human and mouse models of *GRN* haploinsufficiency. Trehalose was verified to function by activating autophagy pathways, enhancing lysosomal functions, and upregulating *GRN* expression. The effects of trehalose were independent of the transcription factor EB (TFEB). Mice treated with trehalose showed increased *GRN* mRNA and PGRN protein levels in their brains. One concern with trehalose, however, was that it may disturb the gut microbiome [134,135].

Benzoxazole-derivatives (e.g., C40, C127, A21, or A41) and blood-brain-barrier-penetrant small molecules upregulated the *GRN* transcription in mice and human cell lines with *GRN* haploinsufficiency [174]. The lysosomal functions were also improved after the A41 treatment [174]. Another small molecule, Ezeprogind, for targeting PGRN, was suggested to be a promising candidate for treating different neurodegenerative diseases. This drug modulated the PGRN–Prosaposin (PSAP) axis, increased progranulin levels in the brain, and improved lysosomal function. Next, AZP2006 showed reduced Tau phosphorylations and neuroinflammations, which showed good tolerance in the Phase 2a phase in patients with Progressive Supranuclear Palsy, and was suggested as a candidate for AD treatment. However, it has not been tested in FTD patients yet [175,176]. A recent study analyzed Honokiol (HNK), a small-molecule polyphenolic compound, in FTD mouse models. HNK was suggested to function as an antitumor agent and restore GRN expression in mice with *GRN* haploinsufficiency. HNK may upregulate the transcription and translation of PGRN protein, which could increase the lysosomal trafficking of PGRN protein through the SORT1-mediated pathway. This data suggested that treatment with HNK could be a novel and safe approach in stimulating PGRN expression. However, further studies on the exact mechanism of HNK in *GRN* haploinsufficiency were essential [140,141]. Table 2 presents examples of therapeutic candidates targeting *GRN* haploinsufficiency.

Anti-SORT1 monoclonal antibodies, especially K1-67, could also be potential molecules to target the PGRN-SORT1 axis. K1-67 increased PGRN levels in both mouse and human cell lines, while other antibodies, such as K1-19 and K1-32, only stimulated *GRN* expression in human cell lines. PGRN levels in plasma, interstitial fluid, and CSF were increased in treated cell lines with K1-67. Anti-SORT1 successfully upregulated *GRN* expressions in mouse models [137]. However, TMEM106B inhibition failed to protect against *GRN* haploinsufficiency-related phenotypes in mouse models, though some lysosomal phenotypes, including beta-glucuronidase activity, did improve [113]. A monoclonal antibody, Latozinemab (AL001), was also suggested to inhibit the interactions between the PGRN and SORT1, which would increase PGRN levels in plasma and CSF in mouse and primate models. Mice with *GRN* haploinsufficiency presented improved behavior after AL001 treatment. In the Phase 1 clinical trials, reduced SORT1 and increased PGRN levels were presented [177].

Additional studies suggested that inhibiting TREM2 activity in *GRN* haploinsufficiency resulted in microglial hyperactivation. Antagonistic TREM2 antibodies were suggested to reduce TREM2 signaling by enhancing TREM2 shedding. *GRN* haploinsufficient cell lines and mice treated with TREM2 antibodies showed lower microglial activity and phagocytosis. However, lysosomal dysfunctions and impaired lipid and glucose metabolism did not improve with TREM2 antibody treatment. Synaptic loss and NFL levels were lower in the CSF of *GRN* haploinsufficient mice treated with TREM2 antibodies. This study suggested that TREM2 inhibition could reveal neuroprotective effects by inhibiting microglial hyperactivation. TREM2 may have a dual role in microglial functions, as TREM2 inhibition could also result in neurotoxicity. Also, TREM2 antibodies alone may not be a definitive treatment for *GRN* haploinsufficiency, as they did not prevent lysosomal dysfunctions. The significance of TREM2 antibodies in *GRN* haploinsufficiency remained unclear, but they may be a promising part of combination treatment for FTD [115,138].

The CRISPR-Cas9 system revealed significant promise in repairing genetic errors as a focused treatment for various diseases [178,179]. Gene correction using CRISPR-Cas9 has shown promising results in the treatment of muscular dystrophy [180] and ALS [181]. CRISPR-Cas9 may also be a promising approach in fixing haploinsufficiency [182,183]. To date, no study has been performed to correct *GRN* haploinsufficiency; however, recently, applying CRISPR-Cas9 for haploinsufficiency correction has been suggested as a promising treatment for *GRN* mutations [181,184,185,196,197,198]. Furthermore, a mouse study was reported, where CRISPR-Cas correction was performed to correct a pathogenic mutation (Pro301Ser) in *MAPT*, which could also be a causative factor for FTD. The levels of soluble and insoluble Tau proteins were reduced in the mouse models [197]. Also, iPSC cells from patients with C9orf72 repeat expansion were targeted with the CRISPR-Cas13 system. This study revealed reduced levels of endogenous sense and antisense repeat RNAs and dipeptide repeats in the cell lines. Also, CRISPR-Cas13 correction was protective against excitotoxicity [199]. Studying genetic correction using CRISPR-Cas9 or CRISPR-Cas13 in cases of *GRN* mutations is also a promising new avenue of research.

One of the issues with stimulating *GRN* expressions was its potential oncogenic properties [121,142]. Furthermore, a recent study revealed that, in addition to *GRN* haploinsufficiency, the overexpression of normal *GRN* could impact the FTD-like abnormalities in mice through gain-of-function mechanisms, including reduced lifespan, behavioral dysfunctions, cognitive decline, gliosis, and lysosomal dysfunctions. In cell cultures, *GRN* overexpression led to cytotoxic effects due to ER stress and increased apoptosis. This data suggested that therapies should consider optimizing mechanisms in regulating *GRN* expressions [143].

## 7. Discussion

The *GRN* gene was verified as a common genetic factor for FTD, resulting in disease phenotypes through haploinsufficiency. *GRN* haploinsufficiency resulted in reduced protein levels, resulting in disrupted lysosomal homeostasis, impaired neuronal survival pathways, and dysregulated inflammatory responses [96,144]. Several mutations, including Arg493Ter or Ser116Ter, were associated with loss-of-function mechanisms from the reduced levels of full-length PGRN protein in biological fluids, such as plasma or CSF [145,146].

*GRN* deficiency caused lysosomal dysfunction, characterized by the accumulation of toxic substrates and altered protease activity [147]. Additionally, the loss of *GRN* contributed to heightened neuroinflammation through altered microglial and immune cell states, further exacerbating neurodegenerative processes [148]. Reduced *GRN* expressions disrupted microglial homeostasis, since it could upregulate the levels of pro-inflammatory factors and reduce clearance of toxic aggregates (e.g., TDP43 aggregates) [84]. Mouse models with *GRN* haploinsufficiency presented abnormal social behavior issues, including social dominance dysfunctions, which could be caused by abnormal dendritic arborization and dendritic spine density [103,117].

Several cell and animal models demonstrated the *GRN* haploinsufficiency. Studies on iPSCs or iMGs were performed to understand the mechanisms of haploinsufficiency, which suggested its replicated impairments in several phenotypes of FTD [95,98,149]. These cell studies were promising because they accurately demonstrated the potential effects and pathological hallmarks of *GRN* haploinsufficiency. Using human-derived cell models from FTD-GRN patients provided a precise study of mutation-specific phenotypes and biomarker correlations (e.g., lower plasma PGRN, elevated CSF neurofilament light chain, and inflammatory markers), which may be challenging to evaluate comprehensively in animal models. These models provided a platform for testing therapeutic interventions targeting the pathological hallmarks of FTD with *GRN* haploinsufficiency [84,92]. However, cell models had several limitations. These models may not mimic proper brain anatomy, functions, or neural connectivity, since they were isolated from specific cell types [84,148]. Other limitations of cell models included the requirement for complex protocols and devices to model the exact functions of neurons and microglia, which may not be available to all research groups [84]. Additionally, cell cultures had limited lifespans, and phenotypic stability could be unstable. They may not clearly reflect chronic or age-related cellular changes and may fail to present age-related and long-term disease progression [92]. Furthermore, cell models may be limited to microglia and neurons and may not effectively model the intercellular pathways involved in disease progression [148].

Animal models, especially rodent models, successfully presented FTD-related changes [116,150,151]. Animal models could be effective in testing gene therapy, protein expression replacement, or optimizing medicine dosage [152,153,154]. A challenge with GRN mice was that homozygous and heterozygous *GRN* knockout mice failed to clearly present human FTD-like phenotypes, since heterozygous *GRN* knockout (GRN+/−) generally showed a low degree of neuropathology, while homozygous *GRN* knockout mice displayed more severe phenotypes, which were rarely observed in FTD patients. This discrepancy between mild phenotypes in heterozygous mice and severe neurodegeneration in human haploinsufficiency patients suggested that there may be species-specific differences in *GRN* biology and/or functions in mice in comparison to humans [108,153,155,156].

Patient stratification would also be important in designing clinical trials of complex diseases, including FTD. FTD haploinsufficiency was verified as one of the main causative factors for FTD, where the reduced progranulin levels would drive the loss-of-function mechanisms [186]. Better aspects of patient stratification should be the utilization of biomarker analysis, since the reduced PGRN levels in biological fluids could predict the pathogenic *GRN* mutations in the asymptomatic individuals and could be a non-invasive method in monitoring disease progression [187]. In case of *GRN*-haploinsufficiency associated FTD, clinical phenotypes may be diverse, including language-memory and behavioral dysfunctions, but motor impairment may also be possible, especially in later disease stages. Understanding the specific phenotypes of diseases, including FTD, could allow better targeted therapeutic approaches. Complex diseases, like FTD with various clinical manifestations, may require different therapeutic strategies, since the variability of phenotypes and progression may require careful stratification [57,187]. Imaging markers of FTD, including alterations in white matter and in brain metabolisms, could also support patient stratification. Imaging could be helpful in identifying disease-related regions and facilitating early disease diagnosis and intervention [188,189]. The presence of genetic modifiers, such as TMRM106B in the case of *GRN* haploinsufficiency, should also be important for patient stratification, as patients with potential disease modifiers may suggest alternative therapies [190]. Patient stratification could also be critical in monitoring potential treatment responses in *GRN*-associated FTD patients, as well as in optimizing treatment strategies for these patients. The heterogeneity of *GRN*-associated FTD suggested that treatment efficacy may vary across different patient subgroups. By stratifying patients, clinical trials could provide a more accurate assessment of the effectiveness of new drugs, such as progranulin-stimulating therapies, and tailor interventions to specific patient profiles [125,126,127,128,129,130,131,132,133,134,135,136,137,138,139,140,141,142,143,144,145,146,147,148,149,150,151,152,153,154,155,156,157]. Disease prediction in presymptomatic stages would be essential, as therapies and prevention strategies may be more effective at the early stages of disease progression. However, ethical challenges may arise in the preclinical stages of patient research. Informed consent must be obtained from individuals along with genetic counseling. This would be essential to ensure they were fully apprised of the risks, efficacy, benefits, and alternatives of testing, as well as the potential severity, variability, and treatment strategies for the condition [191,192]. Disease diagnosis in the presymptomatic stage could result in psychological consequences in the patients or disturbance in family dynamics [193]. Research on presymptomatic individuals should follow ethical guidelines and consider the long-term effects, as well as possible unintended consequences, of therapeutic strategies, especially those involving gene therapy [194,195,196].

Currently, no exact therapy has been suggested for FTD. While several strategies showed promise for increasing *GRN* expression or supplementing PGRN protein, a critical evaluation revealed distinct advantages and limitations among these different therapeutic approaches. For example, the AAV-based PR006 gene therapy offered the potential for long-term effects, as the viral vector was designed to provide sustained GRN expression after a single administration—a major benefit over chronic dosing regimens. On the other hand, challenges of AAV-based gene therapies included the potential for long-term immunological responses or the difficulty in reversing unintended side effects. Furthermore, the complex of *GRN* delivery could require invasive procedures [122,124,125,126,127,128,129,130,131,132,133,134,135,136,137,138,139,140,141,142,143,144,145,146,147,148,149,150,151,152,153,154,155,156,157]. Other methods, including small molecule interventions (e.g., small-molecule PGRN-specific binders), could offer simpler oral administration, which would require continuous treatment to maintain therapeutic levels, thereby presenting different compliance and needs of considerations from the systemic exposure [122,124,125,126,127,128,129,130,131,132,133,134,135,136,137,138,139,140,141,142,143,144,145,146,147,148,149,150,151,152,153,154,155,156,157]. The precise targeting of ASOs could also offer alternative treatment, but may require repeated administration, and their delivery may be challenging. Epigenetic modulators and various small molecules could provide alternative strategies for *GRN*-related therapies, since they could enhance gene expression, reduce lysosomal degradation, or inhibit SORT1. Still, these studies have not yet been approved in clinical trials, and they may require repeated administration. Further studies are needed on their long-term and side effects [121,122,123,124,125,126,127,128,129,130,131,132,133,134,135,136,137,138,139,140,141,142,143,144,145,146,147,148,149,150,151,152,153,154,155,156,157]. Effective GRN therapy faces the challenges of designing delivery systems that can efficiently cross the BBB and distribute throughout the central nervous system. Consequently, the choice of therapeutic strategy must consider the disease stage, patient specificity, and the availability of continuously improving, safer delivery technologies. Approaches targeting the SORT1-PGRN axis with monoclonal antibodies, such as Latozinemab, may offer high target specificity and a potentially less frequent dosing schedule than small molecules, but different challenges, related to immunogenicity and brain penetration, would be present. Similarly, while PTC compounds could be promising in restoring full-length protein, current iterations may face significant difficulties regarding toxicity and bioavailability. Future efforts must, therefore, prioritize head-to-head comparisons, refining delivery mechanisms, and exploring synergistic combination therapies that could collectively address the complex pathology of *GRN*-FTD, while carefully balancing the benefits against the unique risks and practicalities of each approach should be considered [121,122,123,124,125,126,127,128,129,130,131,132,133,134,135,136,137,138,139,140,141,142,143,144,145,146,147,148,149,150,151,152,153,154,155,156,157,158,159,160]. Taken together, monitoring the long-term effects of therapies for *GRN* haploinsufficiency in pre-clinical and clinical trials should be crucial [164,165]. One of the reasons could be the chronic nature of neurodegenerative diseases, including FTD, which could require long-term therapeutic interventions [122,124,125]. Lastly, the long-term effects of therapeutic strategies, including gene therapies, on the risk for off-target or virus reactivation, must be assessed carefully, especially when the long-term safety of gene therapy remains unknown [197].

One of the issues with gene therapies is that *GRN* was categorized as an oncogene; its overexpression may promote tumor progression [19,20]. Additionally, therapeutic candidates may experience side effects; for example, PTC readthrough-related therapies can cause neurotoxicity. Gene therapies may carry the risk of off-target effects [129,130,158]. Additionally, *GRN* overexpression could result in neurotoxicity through gain-of-function mechanisms, which should not be overlooked. Additional research would be needed to balance the beneficial and damaging effects of *GRN*-targeting therapies. It will also be important to optimize the precise dosage, drug delivery, and strategies for monitoring FTD patients [158,159].

## Figures and Tables

**Figure 1 ijms-26-09960-f001:**
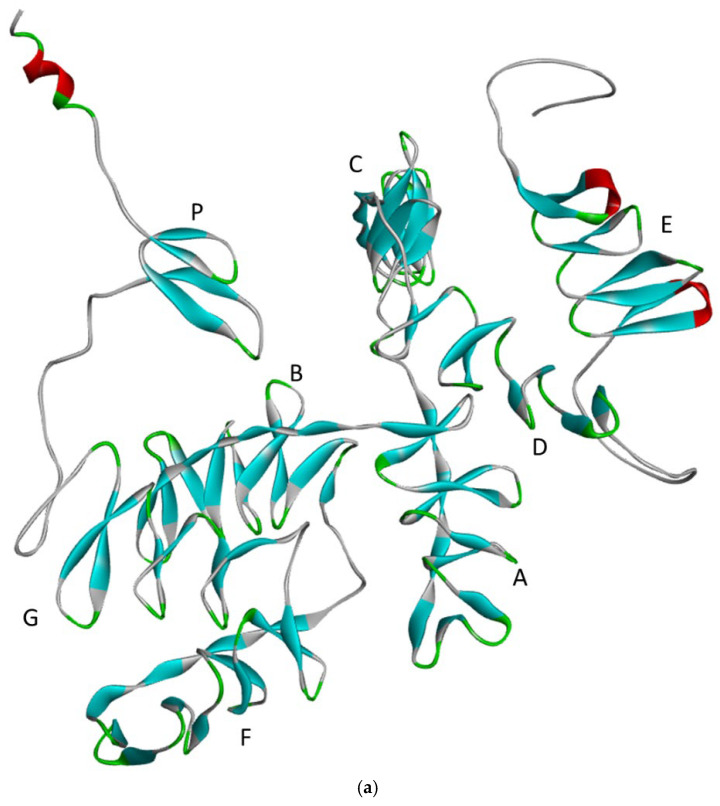
(**a**) 3D structure of progranulin protein, generated by AlphaFold Colab (https://colab.research.google.com/github/sokrypton/ColabFold/blob/main/AlphaFold2.ipynb, accessed on 8 August 2025). The blue means beta sheet, red means alpha helix -or helix like motif, the white means loops, while the green areas mean kinks or turns in the loops. (**b**) Semantic figure of PGRN protein processing and trafficking. PGRN could be cleaved in the extra-and intracellular regions too by different enzymes. PGRN protein could enter to the intracellular endosomes by various receptors (e.g., sortilin1 or mannose-6-phosphate receptors ), and go through exocytosis via the Golgi system or it could be transported and processed in the lysosomes (e.g., by cysteine proteases or cathepsin L) [10].

**Figure 2 ijms-26-09960-f002:**
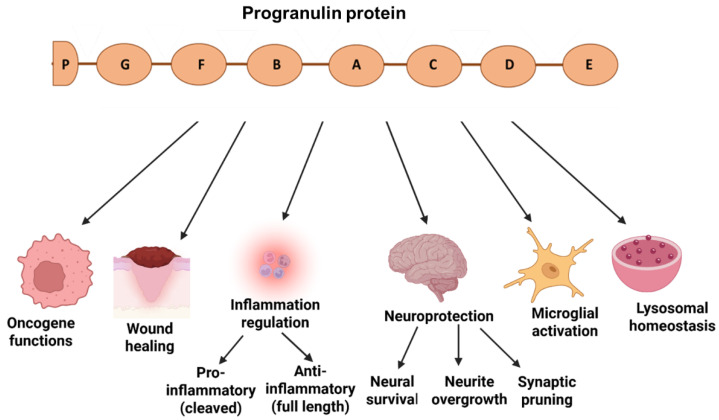
Diverse functions of the PGRN protein, which could impact cell growth, wound healing, inflammatory processes, or neuroprotective mechanisms, including oncogenesis, wound healing, inflammation, neuroprotection, microglial activation or lysosomal homeostasis.

**Figure 3 ijms-26-09960-f003:**
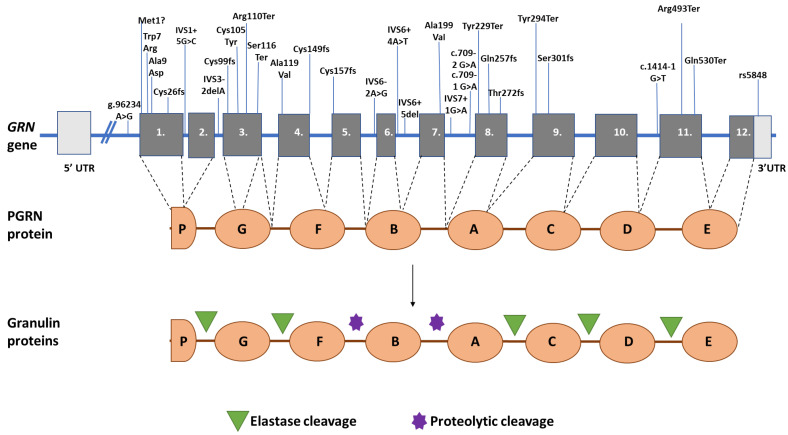
Location of *GRN* mutations, associated with haploinsufficiency in the *GRN* gene and PGRN protein, and location of cleavage sites in GRN proteins. ?: This means a mutation in the START codon. The bright grey means non-coding exon, while the grey means coding exons. Numbers mean the exons of GRN gene. Letters mean the different domains in PGRN protein.

**Figure 4 ijms-26-09960-f004:**
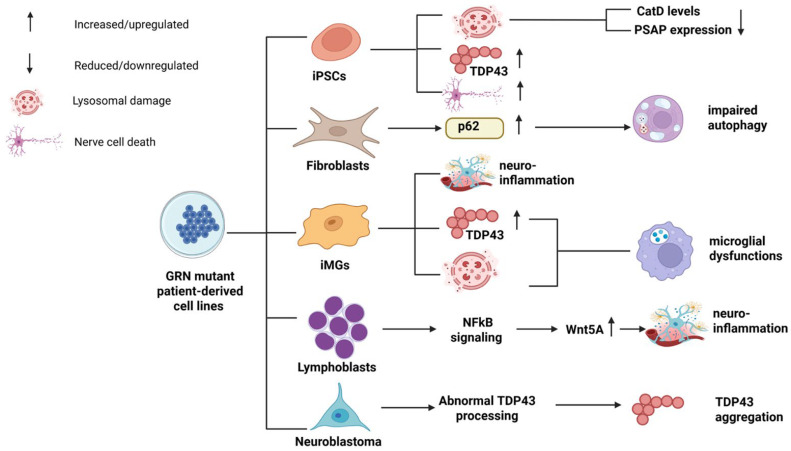
Summary of mechanisms associated with *GRN* haploinsufficiency, based on patient-derived cell models or cell lines with *GRN* suppression. Cell models presented different abnormalities, including lysosomal impairment, TDP43 aggregation, or neuroinflammation.

**Table 1 ijms-26-09960-t001:** Examples of associated *GRN* variants with haploinsufficiency. AOO means age of onset, PNFA means progressive non-fluent aphasia.

Type of Mutation	Mutation	FTD Phenotypes	AOO	Family History	Imaging	Reference
Start codon loss	c.1A > G, p.Met1?	Language or behavioral disorders, depression, bulimia	57	+	NA	[41]
Splice site	g. 96234 A > G	apathy, loss of interests, delusions, attention deficits, and language impairment	69	+	Asymmetric cerebral atrophy in the right hemisphere, frontal temporal lobe	[52]
	IVS1 + 5G > C	Primary progressive aphasia or behavioral FTD	45–70	+/−	TDP43 positivity	[53,54]
	IVS3−2delA	Language and memory dysfunctions	63	+	Brain glucose hypometabolism in several brain areas	[55]
	IVS6−2A > G	Language, behavioral, and memory impairment	50–68	+	Ubiquitin-positive FTD	[56]
	IVS6 + 5_8delGTGA	Apathy, social withdrawal, depression, and language impairment	54	+	Mild diffuse cortical atrophy	[57]
	708 + 4A > T	Apathy, language impairment, and executive dysfunctions	69	+	Diffuse cortical atrophy in the left hemisphere hypometabolism involving the frontal, parietal, and temporal cortices	[58]
	IVS7 + 1G- > A	Corticobasal syndrome	62	+	Asymmetric hemispheric cortical atrophy and ventricular dilatation	[59]
	c.709−1 G > A	bvFTD, PNFA, AD-or PD-like symptoms	42–71	+/−	Gray matter loss in the frontal and parietal lobes, parietal atrophy	[60,61]
	c.709−2 A > T	Behavioral changes, apathy, disinhibition, aggression, disinhibition, ADS	43–80	+	Bilateral frontal atrophy	[62]
	c.1414−1G > T	bvFTD, apathy, perseverative behavior, hyperorality, aphasia, AD-like symptoms	59–69	−/+	Hypoperfusion in the frontotemporal lobes/ parietal cortices	[38]
Frameshift	Cys26Serfs*28	Language impairment, minor motor impairment	63	NA	Left-sided frontotemporal atrophy	[63]
	Cys99Profs*15	Motor, language, and cognitive dysfunctions, behavioral issues	57–63	+	Cortical atrophy	[64]
	Cys149fs*10	Language deficits, apathy, emotional lability, anhedonia, depressive mood, and anosognosia	60s	+	Asymmetric frontal 114 and temporal cortical atrophy	[65]
	Cys157Lysfs*97	bvFTD nonfluent/agrammatic variant of primary progressive aphasia	50s	+/−	Hypometabolism in the left medial temporal cortex, frontal cortex, and posterior cingulate	[66]
	Gln257fs	Memory dysfunctions, language impairment	67	−	Asymmetrical, right-dominant frontoparietal atrophy	[67]
	Thr272fs	Behavioral variant FTD	53–63	+/−	Asymmetric atrophy, frontal lobes	[68]
	Ser301Cysfs*61	bvFTD, corticobasal syndrome, PNFA	53–60	+	NA	[40,69]
Stopgain	Arg110Ter	Motor and language impairment, or bvFTD	50–60s	+	Atrophy and hypometabolism in frontal temporal areas	[70]
	Ser116Ter	Typical FTD	57	+	NA	[39]
	Tyr229Ter	Dyspraxia, dysgraphia, dysphasia, hemiparesis, and depression	60	+	Mild atrophy predominantly in parietal and frontal regions, TDP43-positive inclusions	[71]
	Tyr294Ter	PNFA or bvFTD	54–70	+	TDP43 positivity possible, frontal lobes atrophy on the left side	[72,73]
	Arg493Ter	Bv FTD, PPA, memory-and executive impairment	44–69	+/−	Atrophy, hypometabolism, or hypoperfusion in the frontal/frontotemporal lobes	[51,74]
	Gln530Ter	bvFTD	70s	+	NA	[75]
Missense	Trp7Arg	bvFTD, apathy, diet change, impaired attention, language, and memory	53	+	Asymmetrical frontal, temporal, and parietal atrophy and hypometabolism	[43,44]
	Ala9Asp	FTD with behavior and language deficits	52–77	+	Ubiquitin positive, Tau negative, atrophic hippocampus may be possible	[44,76]
	Cys105Tyr	Psychomotor agitation, motor, and memory impairment	70s	+	Hypometabolism in the left frontotemporal lobe	[77]
	Ala119Val	Language and motor impairment, behavioral issues	61–66	+	Bilateral frontal and parietal cortical atrophy	[78]
	Ala199Val	Typical FTD	62	+	NA	[77]

?: This means a mutation in the START codon. *: Used to mark the position of the stop codon.

**Table 2 ijms-26-09960-t002:** Therapeutic candidates targeting the haploinsufficiency in *GRN*.

Therapeutic Approach	Mechanisms	Benefits	Limitations	References
Gene Therapy (PR006, AVB-101, PBFT02)	Delivery of functional GRN via AAV vectors	Long-term expression of GRN Tolerable in patients	Invasive methods needed (e.g., intracranial injection), risk for off-target or abnormal immune response	[122,123,124,125,166,167,168,169,170]
Non-viral Delivery (e.g., lipid nanoparticles, hydrogels)	Non-viral GRN gene/protein delivery	Non-invasive methods, easier to release	Not tested for *GRN* yet; delivery efficiency uncertain	[173]
Epigenetic Modulation (HDAC inhibitor	Reactivate GRN expression by altering chromatin state	Small molecule-based, reversible mechanisms	Limited efficacy (FRM-0334 trial failed); bioavailability issues	[127,128]
Small Molecules (e.g., Amiodarone, Bafilomycin A1)	Enhance GRN expression or lysosomal function	Oral/small-molecule delivery; targets secondary pathways	Off-target effects, variable potency, microbiome, or acidosis issues	[116,133,134,135,136,140,141]
PTC Readthrough Compounds (e.g., G418, gentamicin)	Promote translation through premature stop codons	May restore GRN expression and PGRN protein levels	Toxicity, low bioavailability, and specificity to nonsense mutations	[129,130]
SORT1 Pathway Modulation (e.g., AL001)	Prevent PGRN degradation via SORT1 blockade	PGRN elevation in CSF/plasma; well-tolerated in early trials	Long-term efficacy unknown; possible compensatory effects	[113,177]
TREM2 Modulation (Anti-TREM2 antibodies)	Reduce microglial hyperactivation	Potential neuroprotection, reduced synaptic loss	No improvement in lysosomal dysfunction; possible neurotoxicity	[115]
ASOs (e.g., anti–miR-29b)	Block microRNA-related repression of *GRN*	Allele-independent; adaptable	Requires repeated dosing; off-target risk	[140]
CRISPR-Cas Systems (Cas9, Cas13)	Promising future study in correcting GRN mutation or expression	Potential for permanent correction	Ethical, delivery, and safety challenges; no study on GRN correction yet	[178,179]

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
