# Peer review of "Targeting Granulin Haploinsufficiency in Frontotemporal Dementia: From Genetic Mechanisms to Therapeutics"

_ijms, 2025, doi:10.3390/ijms26209960_

Round 1

Reviewer 1 Report

Comments and Suggestions for Authors

The manuscript provides an extensive review of progranulin (GRN) haploinsufficiency in frontotemporal dementia (FTD), summarizing genetic mechanisms, biomarker development, cellular and animal models, and emerging therapeutic strategies. The topic is highly relevant to both the scientific and clinical communities, and the breadth of coverage is commendable. The figures and tables are informative and contribute to the reader’s understanding of the complex biology. Nevertheless, the review currently reads as a descriptive compilation rather than a critical synthesis, with some issues in organization, insufficient discussion of recent advances, and limited translational perspective. Addressing these concerns would significantly strengthen the manuscript and enhance its contribution to the field.

Major Issues

  1. Several sections, particularly those on biomarkers, patient-derived models, and animal models, are overly descriptive and read as long lists of findings. This makes it difficult for readers to distill key insights. For instance, the biomarker section outlines numerous fluid-based markers but does not clearly prioritize which are most promising for early detection, prognosis, or therapeutic monitoring. I recommend adding concise summaries at the end of each subsection that highlight the main take-home messages and their implications for research or clinical practice.
  2. The manuscript relies heavily on studies published before 2022, with limited incorporation of recent advances. Key areas such as CRISPR/Cas9-mediated GRN correction, nanoparticle-based delivery systems, and novel small molecules targeting GRN pathways are underrepresented. To maintain the timeliness of the review, the authors should integrate findings from the latest publications in Nature Medicine, Neuron, Brain, and other leading journals.
  3. While the therapeutic approaches are well described, the discussion does not sufficiently compare their relative advantages and limitations. For example, gene therapy and antisense oligonucleotides differ substantially in terms of durability of effect, safety profiles, delivery challenges, and cost. Without critical evaluation, the review risks being perceived as a catalogue of methods rather than a scholarly synthesis. I encourage the authors to include comparative tables or narrative sections that highlight these strengths and weaknesses.
  4. The manuscript provides limited insight into the major hurdles in translating these strategies into clinical application. Specific issues such as long-term safety monitoring, efficient blood-brain barrier penetration, patient stratification for clinical trials, and ethical considerations for presymptomatic interventions require more attention. Expanding the discussion on these translational bottlenecks would increase the clinical relevance of the review.
  5. The manuscript contains repetitive statements and wordy phrasing that could be streamlined. Careful editing for conciseness, clarity, and avoidance of redundancy would enhance readability.

Minor Issues

  1. Figures, especially Figures 3 and 4, are text-heavy and difficult to interpret quickly. Simplifying labels, improving contrast, and focusing on essential elements would improve their effectiveness.
  2. Several abbreviations (e.g., AOO, PNFA) are not defined at first use. These should be explained upon initial mention.
  3. The Discussion section contains considerable overlap with earlier sections (e.g., biomarker and animal model descriptions). Condensing repetitive material would improve focus and reduce redundancy.

Author Response

The manuscript provides an extensive review of progranulin (GRN) haploinsufficiency in frontotemporal dementia (FTD), summarizing genetic mechanisms, biomarker development, cellular and animal models, and emerging therapeutic strategies. The topic is highly relevant to both the scientific and clinical communities, and the breadth of coverage is commendable. The figures and tables are informative and contribute to the reader’s understanding of the complex biology. Nevertheless, the review currently reads as a descriptive compilation rather than a critical synthesis, with some issues in organization, insufficient discussion of recent advances, and limited translational perspective. Addressing these concerns would significantly strengthen the manuscript and enhance its contribution to the field.

 Thank you very much for the constructive peer review. We revised the manuscript, regarding your suggestions.

Major Issues

  1. Several sections, particularly those on biomarkers, patient-derived models, and animal models, are overly descriptive and read as long lists of findings. This makes it difficult for readers to distill key insights. For instance, the biomarker section outlines numerous fluid-based markers but does not clearly prioritize which are most promising for early detection, prognosis, or therapeutic monitoring. I recommend adding concise summaries at the end of each subsection that highlight the main take-home messages and their implications for research or clinical practice.

Thank you, we added some take-home messages in the biomarker chapter

“As previously mentioned above, the majority of GRN mutations were associated with loss-of-function mechanisms, leading to reduced plasma PGRN levels in affected patients. Analyzing plasma PGRN in novel GRN carriers became an effective method to evaluate the pathogenicity of a its mutations. Moreover, monitoring PGRN levels in biological flu-ids could be a valuable approach for assessing the efficacy of drug candidates that target-ed GRN haploinsufficiency [79, 80, 81]. Circulating PGRN levels were reduced in serum or CSF in cases of mutations in association with its haploinsufficiency [73,82, 83]. Among these biomarkers, monitoring plasma PGRN levels would be crucial for identifying individuals with probably pathogenic GRN mutations, since the reduced plasma progranulin levels reflected the loss-of function mechanisms [79-81]. Furthermore, plasma PGRN levels suggested as a potential biomarker in predicting disease progression in pre-symptomatic GRN mutation carriers [80]. They may also be important in monitoring the potential drug candidates of GRN haploinsufficiency, since the successful candidates may increase and stabilize the plasma granulin levels [79-83].”

“Another promising biomarker for GRN haploinsufficiency was neurofilament light chain (NfL) protein, which was considered an accurate marker for both neuroinflammatory and neurodegenerative changes. The elevated levels of NfL could be useful in the ear-ly detection of GRN-haploinsufficiency-related neurodegenerative damage [84]. Addition-ally, CSF NfL levels correlated with disease severity and with hypometabolism in affected brain regions [85]. Hence, CSF NfL could be useful in monitoring the FTD drug candidates, since their reduced levels may be associated with a positive therapeutic outcome, resulting in slowdown in neurodegeneration [84,85].”

Plasma analysis of patients with GRN haploinsufficiency-associated mutations also revealed the elevated levels of peripheral biomarkers related to macrophage-mediated innate immunity, such as plasma sCD163 or CCL18, which were higher in GRN mutation carriers than controls. Plasma LBP may serve as a useful marker for disease severity in GRN mutation carriers, as its levels correlated with frontal white matter integrity [86]. A separate study indicated that FTD patients with GRN haploinsufficiency had higher serum concentrations of C-peptide and resistin. Another biomarker Ghrelin was found to be increased in both pre-symptomatic GRN mutation carriers and FTD patients, suggesting that metabolic biomarker alterations may be associated with FTD progressions [87]. Research into the impact of GRN haploinsufficiency on brain sphingolipid enzymes revealed the reduced beta-glucocerebrosidase activity in the inferior frontal gyrus of GRN carrier FTD patients. Specifically, mature enzyme levels were lower, and insoluble and glycosylated beta-glucocerebrosidase was activated in their neurons. These findings, corroborated in GRN knockout mice, suggested that GRN haploinsufficiency may be linked to impaired beta-glucocerebrosidase processing [88]. Elevated peripheral inflammatory markers (e.g., sCD163, CCL18) and metabolic biomarkers (e.g., C-peptide, resistin, ghrelin) in GRN carriers would be useful markers in monitoring the disease severity and metabolic alterations, potentially predicting a more aggressive disease course. This suggested a broader panel of markers for better informed prognostic evaluations. Changes in the levels of peripheral inflammatory markers or metabolic markers may also be useful in monitoring the efficacy of therapeutic candidates for GRN haploinsufficiency [86-88].

  1. The manuscript relies heavily on studies published before 2022, with limited incorporation of recent advances. Key areas such as CRISPR/Cas9-mediated GRN correction, nanoparticle-based delivery systems, and novel small molecules targeting GRN pathways are underrepresented. To maintain the timeliness of the review, the authors should integrate findings from the latest publications in Nature Medicine, Neuron, Brain, and other leading journals.

Thank you, we added several more up-to date studies in this chapter.

The AVB-101 was also an AAV (AAV9)-based therapy, which started its clinical Trial1/2 in individuals with GRN mutations and FTD therapy. However, no information was available on its effect on PGRN levels or safety. Preclinical studies in primates revealed the increased PGRN levels in the brain, cardiovascular system and liver after AVB-101 treatment [166, 167]. PBFT02 was also an AAV-based gene therapy, currently in Phase1b. The initial results of PBFT02 seemed to be promising, since the tested patients presented elevated CSF progranulin levels at 6 months from the start of the treatment. Also, the therapy seemed to be well tolerated in patients [168,169, 170]. The benefit of AAV vector-related gene therapies was the long-term its expressions in the neurons. Preclinical data and early clinical data presented the elevated gene expressions. In addition, AAV based methods facilitated to target specific cell types in or brain regions, and they were well tolerated in preclinical and clinical studies. However, AAV based therapies had several limitations. One of the issues was the invasive procedures of intracranial injections to efficiently target the brain. Other issue could be that AAV induced immune responses, which may need immunosuppression procedures. Furthermore, the risk for off-target effects needed to be assessed [124,125].”

“Non-viral vectors, including lipid nanoparticles, may also be promising in gene de-livery. Up to date, no study was performed on GRN delivery by lipid nanoparticles, but it may be a safe an effective method in studying haploinsufficiency [171]. Additionally, the nanoparticles with enhanced stability and control the release of drugs should be essential for the drug delivery, especially targeting specific sites [172]. A thermo-sensitive hydrogel (Pluronic F-127) was successfully used to deliver progranulin and enhanced the healing on corneal injury, which inhibited the inflammation (e.g. by suppressing NF-κB, PI3K/Akt or Wnt/β-catenin signaling) and stimulated the axonal regeneration [173].”

Benzoxazole-derivatives (e. g. C40, C127, A21 or A41) and blood-brain-barrier-penetrant small molecules upregulated the GRN transcription in mice and human cell lines with GRN haploinsufficiency [174]. The lysosomal functions were also improved after the A41 treatment. [174]. Another small molecule, Ezeprogind, of targeting PGRN was suggested to be promising candidate in different neurodegenerative diseases. This drug modulated the PGRN- Prosaposin (PSAP) axis, increased the progranulin levels in the brain, and improved the lysosomal functions. Next, AZP2006 showed the reduced Tau phosphorylations and neuroinflammations, which showed good tolerance in Phase 2a phase in patients with Progressive Supranuclear Palsy, and was suggested as the candidate for AD treatment. However, it was not tested in FTD patients, yet [175, 176]”

“A monoclonal antibody Latozinemab (AL001) was also suggested to inhibit the interactions between the PGRN and SORT1, which would increase PGRN levels in plasma and CSF in mouse and primate models. Mice with GRN haploinsufficiency presented the improved behavior after AL001 treatment. In the Phase-1 clinical trials presented the reduced SORT1 and increased PGRN levels [177].”

The CRISPR-Cas9 system revealed significant promises in repairing genetic errors, as the focused treatment for various diseases [178, 179]. Gene correction CRISPR-Cas9 showed promising results in the treatment for muscular dystrophy [180] or ALS [181]. CRISPR-Cas9 may also be a promising approach in fixing haploinsufficiency [182, 183]. Up to date, no study was performed in correcting GRN haploinsufficiency, but the recently, applying CRISPR-Cas9 for haploinsufficiency was suggested as the promising treatment in correcting GRN mutations [181, 184, 185, 197, 198].  Furthermore, a mouse study was reported, where CRISPR-CAS9 correction was performed to correct a pathogenic mutation (Pro301Ser) in MAPT, which could also be a causative factor for FTD. The levels of soluble and insoluble Tau proteins were reduced in the mouse models [197]. Also, iPSC cells from patients with C9orf72 repeat expansion were targeted with CRISPR-Cas13 system. This study revealed reduced endogenous sense and antisense repeat RNAs and dipeptide re-peat levels in the cell lines. Also, CRISPR-Cas13 correction was protective against excitotoxicity. [199] Studying genetic correction by CRISPR-CAS9 or CRISPR-Cas13 in the case of GRN mutations should also be a promising new avenue of research.”

  1. While the therapeutic approaches are well described, the discussion does not sufficiently compare their relative advantages and limitations. For example, gene therapy and antisense oligonucleotides differ substantially in terms of durability of effect, safety profiles, delivery challenges, and cost. Without critical evaluation, the review risks being perceived as a catalogue of methods rather than a scholarly synthesis. I encourage the authors to include comparative tables or narrative sections that highlight these strengths and weaknesses.

Thank you, we tried to improve these issues in the discussion

“Currently, there are no therapies are available against FTD. However, several strategies showed promise for increasing GRN expression or supplementing PGRN protein, a critical evaluation revealed different advantages and limitations among the different therapeutic strategies. For example, the gene therapies e. g. AAV-based PR006 could offer the potential for long-term effects, since viral vector may lead to sustained GRN expression after a single administration, which could be beneficial, compared to approaches requiring chronic dosing. However, AAV-based gene therapies may also have challenges, including the potential for long-term immunological responses or the difficulty in reversing unintended side effects. Furthermore, the GRN delivery may be complex, and it could require invasive procedures. [122, 124, 125-157]. However, other methods, including small molecule interventions (e.g. small-molecule PGRN-specific binders), could offer simpler oral administration, but they may require continuous treatment to maintain therapeutic levels, thereby presenting different compliance and systemic exposure considerations. [122, 124, 125-157]. ASOs could also offer precise targeting but they also may require repeated administration and their delivery may be challenging. Epigenetic modulators and various small molecules could provide alternative strategies of GRN-related therapies, since they could enhance gene expressions, reduce lysosomal degradations, or inhibit SORT1. However, these studies were not approved in clinical trials yet, and they may require repeated administration. Further studies should be needed on their long-term effects and side effects [121-157]. Ultimately, the choice of therapeutic strategy should depend on a combination of disease stage, patient specificity, and the ongoing development of safer and more efficient delivery systems. Approaches targeting the SORT1-PGRN axis with monoclonal antibodies, e. g.  Latozinemab, may offer high target specificity and a potentially less frequent dosing schedule than small molecules, but they may have different challenges, related to immunogenicity and brain penetration. Similarly, while PTC readthrough compounds could be promising in restoring full-length protein, current iterations may face significant difficulties regarding toxicity and bioavailability. Future efforts must therefore prioritize head-to-head comparisons, refining delivery mechanisms, and exploring synergistic combination therapies that could collectively address the complex pathology of GRN-FTD while carefully balancing the benefits against the unique risks and practicalities of each approach [121-160].’

  1. The manuscript provides limited insight into the major hurdles in translating these strategies into clinical application. Specific issues such as long-term safety monitoring, efficient blood-brain barrier penetration, patient stratification for clinical trials, and ethical considerations for presymptomatic interventions require more attention. Expanding the discussion on these translational bottlenecks would increase the clinical relevance of the review.

Thank you discussed these issues.

On long-term safety:

“Taken together, monitoring the long-term effects of therapies for GRN haploinsufficiency in pre-clinical and clinical trials should be crucial [165]. One of the reasons could be the chronic nature of neurodegenerative diseases, including FTD, which could require long term therapeutic interventions [122, 124,125]. Lastly, the long-term effects of therapeutic strategies, including gene therapies on the risk for off-target or virus reactivation, must be assessed carefully, especially when the long-term safety of gene therapy remained un-known. [197].”

 BBB penetration and long-term safety monitoring in the discussion:

Currently, no exact therapy was suggested for FTD. While several strategies showed promise for increasing GRN expression or supplementing PGRN protein, a critical evaluation revealed distinct advantages and limitations among these different therapeutic approaches. For example, The AAV-based PR006 gene therapy offered the potential for long-term effects, since the viral vector was designed to provide sustained GRN expressions after a single administration, a major benefit over chronic dosing regimens. On the other hand, challenges of AAV-based gene therapies included the potential for long-term immunological responses or the difficulty in reversing unintended side effects. Furthermore, the complex of GRN delivery could require invasive procedures. [122, 124, 125-157]. Other methods, including small molecule interventions (e.g. small-molecule PGRN-specific binders), could offer simpler oral administration, which would require continuous treatment to maintain therapeutic levels, thereby presenting different compliance and needs of considerations from the systemic exposure [122, 124, 125-157]. The precise targeting of ASOs could also offer an alternative treatment, but may require repeated administration and their delivery may be challenging. Epigenetic modulators and various small molecules could provide alternative strategies of GRN-related therapies, since they could enhance gene expressions, reduce lysosomal degradations, or inhibit SORT1. Still, these studies were not approved in clinical trials yet, and they may require repeated administrations. Further studies should be needed on their long-term and side effects [121-157]. Effective GRN therapy faces the challenges of designing delivery systems that can efficiently cross BBB and the distributions throughout the central nervous system. Consequently, the choice of therapeutic strategy must consider the disease stage, patient specificity, and the availability of continuously improving, safer delivery technologies. Approaches targeting the SORT1-PGRN axis with monoclonal antibodies, such as Latozinemab, may offer high target specificity and a potentially less frequent dosing schedule than small molecules, but different challenges, related to immunogenicity and brain penetration, would be present. Similarly, while PTC compounds could be promising in restoring full-length protein, current iterations may face significant difficulties regarding toxicity and bioavailability. Future efforts must, therefore, prioritize head-to-head comparisons, refining delivery mechanisms, and exploring synergistic combination therapies that could collectively address the complex pathology of GRN-FTD, while carefully balancing the benefits against the unique risks and practicalities of each approach should be considered [121-160].”

Patient stratification:

Patient stratification would also be important in designing clinical trials of complex diseases, including FTD. FTD haploinsufficiency was verified as one of the main causative factors for FTD, where the reduced progranulin levels would drive the loss-of-function mechanisms [186]. Better aspects of patient stratification should be the utilization of biomarker analysis, since the reduced PGRN levels in biological fluids could predict the pathogenic GRN mutations in the asymptomatic individuals and could be a non-invasive method in monitoring disease progression [187]. In case of GRN-haploinsufficiency associated FTD, clinical phenotypes may be diverse, including language-memory and behavioral dysfunctions, but motor impairment may also be possible, especially in later disease stages. Understanding the specific phenotypes of diseases, including FTD, could allow better targeted therapeutic approaches. Complex diseases, like FTD with various clinical manifestations, may require different therapeutic strategies, since the variability of phenotypes and progression may require careful stratification [57, 187]. Imaging markers of FTD, including alterations in white matter and in brain metabolisms, could also support the patient stratification. Imaging could be helpful in identifying the different disease related region, and it could also facilitate the early disease diagnosis and intervention [188, 189]. The presence of genetic modifiers, such as TMRM106B in case of GRN haploinsufficiency, should also be important for patient stratification, since the patients with potential disease modifiers may suggest alternative therapy [190]. Patient stratification could be also critical in monitoring the potential treatment responses in the GRN associated FTD patients, as well as in optimizing the treatment strategy for the patients. The heterogeneity of GRN associated FTD suggested that treatment efficacy may vary across different patient subgroups. By stratifying patients, clinical trials could have better assessment of their effectiveness of new drugs, such as progranulin-stimulating therapies and tailor interventions to specific patient profiles [125-157].

On preclinical

“Disease prediction in presymptomatic stages would be essential, since the therapies and prevention strategies may be more effective at the early stages of the disease progression. However, ethical challenges may be considered in the preclinical stages of patients. In-formed consent must be obtained from individuals along with genetic counseling. This would be essential to ensure they were fully apprised of the risks, efficacy, benefits, and alternatives of testing, as well as the potential severity, variability, and treatment strategies for the condition. [191, 192]. Disease diagnosis in presymptomatic stage could result in psychological consequences in the patients or disturbance in family dynamics [193]. Re-search on presymptomatic individuals should follow ethical guidelines and consider the long-term effects along with possible unintended consequences from the therapeutic strategies, especially from the gene therapy [195,196].”

  1. The manuscript contains repetitive statements and wordy phrasing that could be streamlined. Careful editing for conciseness, clarity, and avoidance of redundancy would enhance readability.

Thank you, we tried to reduce the repetitive statements 

Minor Issues

  1. Figures, especially Figures 3 and 4, are text-heavy and difficult to interpret quickly. Simplifying labels, improving contrast, and focusing on essential elements would improve their effectiveness.

Thank you, we tried to improve the figures

  1. Several abbreviations (e.g., AOO, PNFA) are not defined at first use. These should be explained upon initial mention.

Thank you, we added the definition: AOO means age of onset, PNFA means progressive non-fluent aphasia

  1. The Discussion section contains considerable overlap with earlier sections (e.g., biomarker and animal model descriptions). Condensing repetitive material would improve focus and reduce redundancy.

Thank you, we shortened down the repetitive parts in the discussion

Reviewer 2 Report

Comments and Suggestions for Authors

In this review, the authors examined the role of granulin haploinsufficiency in the pathogensis of frontotemporal dementia (FTD) and potential FTD therapeutics strategies that arose from this knowledge.

Overall, the review covered the topic of interest very well,  with clear summary tables and beautifully prepared diagrams that further enhance the understanding of the subject, reflecting the meticulous planning and care in the preparation of the manuscript by the authors. 

Just a few comments:

1) Section 1 (Lines 40-51): considering that the main focus of the review article is Granulin, the detailed discussion of Amyotrophic Lateral Sclerosis (ALS) and Alzheimer's Disease detract the focus away from the main focus considering GRN variants do not play an significant in the pathogenesis of these diseases.

2) Section 2 ( from Line 116): It would be helpful if the authors can mention the prevalence of loss-of-function GRN variants in FTD that resulted in haploinsufficiency to make the logic behind the review clearer

3) Section 3: Typo in title ("and" put in by mistake?)

4)Section 3 (Line 171): Has neurofilament light been tested as a biomarker in clinical trials? Might be helpful to include.

5) Section 4 (from Line 4): considering the clinical presentations of FTD associated with GRN variants are highly heterogeneous, even within the same variant, can authors comment on the variations/ uniformity in terms of the findings from patient derived cell lines?

Author Response

In this review, the authors examined the role of granulin haploinsufficiency in the pathogensis of frontotemporal dementia (FTD) and potential FTD therapeutics strategies that arose from this knowledge.

Overall, the review covered the topic of interest very well, with clear summary tables and beautifully prepared diagrams that further enhance the understanding of the subject, reflecting the meticulous planning and care in the preparation of the manuscript by the authors. 

Thank you for the positive and constructive comments. We revised the manuscript according your suggestion

Just a few comments:

  • Section 1 (Lines 40-51): considering that the main focus of the review article is Granulin, the detailed discussion of Amyotrophic Lateral Sclerosis (ALS) and Alzheimer's Disease detract the focus away from the main focus considering GRN variants do not play a significant in the pathogenesis of these diseases.

Thank you, we reduced this paragraph.

  • Section 2 (from Line 116): It would be helpful if the authors can mention the prevalence of loss-of-function GRN variants in FTD that resulted in haploinsufficiency to make the logic behind the review clearer

Pathogenic mutations in GRN gene were reported in approximately 5-10% of FTD patients, but they could be more frequent (5~20%) in the case of familial FTD [31-34].

  • Section 3: Typo in title ("and" put in by mistake?)

Thank you, we fixed this issue

  • Section 3 (Line 171): Has neurofilament light been tested as a biomarker in clinical trials? Might be helpful to include.

Reviewer 1 asked similar question too, I added some information on the role of NfL in FTD prognosis and drug clinical trials

“Another promising biomarker for GRN haploinsufficiency was neurofilament light chain (NfL) protein, which was considered an accurate marker for both neuroinflamma-tory and neurodegenerative changes. The elevated levels of NfL could be useful in the ear-ly detection of GRN-haploinsufficiency-related neurodegenerative damage [84]. Addition-ally, CSF NfL levels correlated with disease severity and with hypometabolism in affected brain regions [85]. Hence, CSF NfL could be useful in monitoring the FTD drug candidates, since their reduced levels may be associated with a positive therapeutic outcome, resulting in slowdown in neurodegeneration [84,85]

  • Section 4 (from Line 4): considering the clinical presentations of FTD associated with GRN variants are highly heterogeneous, even within the same variant, can authors comment on the variations/ uniformity in terms of the findings from patient derived cell lines?

We added a short paragraph

As mentioned before, patients with pathogenic FTD mutations could present diverse phenotypes, even in case of the same mutations [41-78]. However, in case of patient-derived cell lines, the effects of GRN mutation presented uniformity. The reason of diversity in the patients could be complex between the genetic-or epigenetic factors, and the presence of environmental influences [164]. However, patient-derived cell lines were studied in a controlled environment, since these studies may focus more on the effects on GRN haploinsufficiency [90-92]. 

Round 2

Reviewer 1 Report

Comments and Suggestions for Authors

The authors have responded to the reviewers’ concerns in a serious and constructive manner, and the manuscript has improved significantly. With minor revisions to include a comparative summary table, refine the discussion of models, and polish the language, the paper will be suitable for publication. I recommend acceptance after minor revision.

1.While comparative discussions of therapeutic strategies have been improved, a concise comparative table or schematic summarizing advantages and limitations would make the section more accessible and clinically useful.

2.The patient-derived and animal models sections remain somewhat descriptive; further synthesis is needed to clarify the unique insights each model provides and their implications for translational research.

3.The English language, while improved, still contains occasional awkward phrasing and redundancies. A final round of professional language editing is recommended.

Comments on the Quality of English Language

The English language, while improved, still contains occasional awkward phrasing and redundancies. A final round of professional language editing is recommended.

Author Response

The authors have responded to the reviewers’ concerns in a serious and constructive manner, and the manuscript has improved significantly. With minor revisions to include a comparative summary table, refine the discussion of models, and polish the language, the paper will be suitable for publication. I recommend acceptance after minor revision.

Thank you for the constructive and encouraging comments. We try to revise the manuscript according to your suggestions.

While comparative discussions of therapeutic strategies have been improved, a concise comparative table or schematic summarizing advantages and limitations would make the section more accessible and clinically useful.

Thank you, we replaced the Table 2, and designed a new table on the benefits and limitations of therapeutic strategies

2.The patient-derived and animal models sections remain somewhat descriptive; further synthesis is needed to clarify the unique insights each model provides and their implications for translational research.

Thank you, we added a paragraph each chapter on this issue

Chapter 4:

“Taken together, ​​patient-derived cell lines with various GRN mutations demonstrated different impairments, including TDP43 metabolism, inflammatory dysfunctions, lysosomal abnormalities, or reduced cell survival.  The benefits of these models could be that they could offer a direct human-specific context for understanding GRN haploinsufficiency, providing insights that may be challenging to obtain from animal models [90-110]. Cell models should be essential, since identifying the cellular impairments in the case of GRN haploinsufficiency could provide more opportunities in the case of drug development and testing. The drugs, specifically designed to human physiology could be tested initially in these cell lines. The cell lines could be beneficial in understanding the disease heterogeneity by comparing the responses of the cells from the patients with the same GRN mutations. This should be important in the development of personal medicine development, since understanding the differences in the disease progression and response to treatment may be unique and individual therapeutic strategies should be needed. Furthermore, patient-derived cell lines could provide a scalable and ethical platform high-throughput screening of therapeutic compounds and candidates, before testing them in animal models or patients. The cell lines could also be beneficial in biomarker discovery in the case of granulin haploinsufficiency (e.g. WNT5A), which may be helpful in disease diagnosis and in monitoring disease progression.  [90-101]. The disadvantage of cell lines could be the isolated environment, since they could only focus on the specific cellular effects. Furthermore, the gene-environmental interactions may be difficult to model in the cell lines [90-101].”

Chapter 5:

“As summary, animal models, especially the mouse models, were designed to understand the in vivo effects of GRN haploinsufficiency, since they could provide opportunity to study the complex neural circuits, behavioral phenotypes, and long-term pathological progression that may be difficult to replicate cell systems [102-118]. Mice with GRN haploinsufficiency could presented behavioral issues and circuit impairments in the nerve cells. Also, they presented microgliosis, neuroinflammation or lysosomal dysfunctions [102-118]. Mouse models could have the benefit over animal models that they represent the complex, multi-systemic nature of FTD, including behavioral deficits, neuronal circuit dysfunction, and age-related pathology progression. Animal models could also be essential in testing the therapeutic candidates of FTD, e.g., their safety and their ability to penetrate to the brain, possible side effects or their response to therapies, before moving them to animal models. Also, animal models could be important in the discovery and validation of biomarkers. Furthermore, animal models may be more effective in modeling and understanding gene interactions between GRN and other genes(e.g., with SORT1 or TREM2) and their role in the disease progression. ​Also, studies in animal models may be useful in studying the compensatory mechanisms that could occur in response to GRN deficiency, including increased presynaptic release in hippocampal cultures, which might explain the delayed onset or the various symptoms in the patients [102-118]. However, the disadvantages the animal models could be that they may not exhibit the key neuropathological features of FTD, e.g., TDP43 aggregates. The animal models may not be effective in modeling the human FTD-related disease mechanisms. There may be significant differences between the physiology and anatomy of animals and the humans, which may result in mismatches between the data from human-derived cell lines and animal models.  Ethical concerns may also be an issue in the case of animal models [102-118] .”

3.The English language, while improved, still contains occasional awkward phrasing and redundancies. A final round of professional language editing is recommended.

Thank you, we tried to improve the English of the manuscript